# Highway to Cell: Selection of the Best Cell-Penetrating Peptide to Internalize the CFTR-Stabilizing iCAL36 Peptide

**DOI:** 10.3390/pharmaceutics14040808

**Published:** 2022-04-07

**Authors:** Quentin Seisel, Israpong Lakumpa, Emilie Josse, Eric Vivès, Jessica Varilh, Magali Taulan-Cadars, Prisca Boisguérin

**Affiliations:** 1CRBM, University of Montpellier, CNRS UMR 5237, 34000 Montpellier, France; quentin.seisel@live.fr (Q.S.); israpong.lakumpa@gmail.com (I.L.); 2PhyMedExp, Bâtiment Crastes de Paulet, University of Montpellier, INSERM U1046, CNRS UMR 9214, 34000 Montpellier, France; emilie.josse@inserm.fr (E.J.); eric.vives@umontpellier.fr (E.V.); 3PhyMedExp, Institut Universitaire de Recherche Clinique, University of Montpellier, INSERM U1046, CNRS UMR 9214, 34000 Montpellier, France; jessica.varilh@inserm.fr (J.V.); magali.taulan@inserm.fr (M.T.-C.)

**Keywords:** CFTR-CAL interaction, CFTR stabilizer, cell-penetrating peptide, p.Phe508del mutant, internalization mechanism, cystic fibrosis

## Abstract

Therapeutic peptides have regained interest as they can address unmet medical needs and can be an excellent complement to pharmaceutic small molecules and other macromolecular therapeutics. Over the past decades, correctors and potentiators of the cystic fibrosis transmembrane conductance regulator (CFTR), a chloride ion channel causing cystic fibrosis (CF) when mutated, were developed to reduce the symptoms of the patients. In this context, we have previously designed a CFTR-stabilizing iCAL36 peptide able to further increase the CFTR amount in epithelial cells, thereby resulting in a higher CFTR activity. In the present study, optimization of the peptidyl inhibitor was performed by coupling five different cell-penetrating peptides (CPP), which are Tat, dTat, TatRI (*retro-inverso*), MPG, and Penetratin. Screening of the internalization properties of these CPP-iCAL36 peptides under different conditions (with or without serum or endocytosis inhibitors, etc.) was performed to select TatRI as the optimal CPP for iCAL36 delivery. More importantly, using this TatRI-iCAL36 peptide, we were able to reveal for the first time an additive increase in the CFTR amount in the presence of VX-445/VX-809 compared to VX-445/VX-809 treatment alone. This finding is a significant contribution to the development of CFTR-stabilizing peptides in addition to currently used treatments (small-molecule correctors or potentiators) for CF patients.

## 1. Introduction

The molecular drivers of many basic cellular functions and pathways are protein–protein interactions (PPIs); therefore, they have broadly been used as targets for drug development. Besides small molecules, which are not useful for targeting large or flat PPI binding sites, peptides are increasingly becoming attractive therapeutic candidates. The advantages of therapeutic peptides lie primarily in their high efficacy and low toxicity, but also in the unlimited possibilities of introducing modifications to improve their stability and binding affinity. Accordingly, therapeutic approaches using peptides have become an emerging market in the pharmaceutical industry over the past two decades [1], with more than 60 peptide drugs approved by the Food Drug Administration and many more studied in clinical and preclinical trials [2].

Within the context of cystic fibrosis (CF), an inherited recessive autosomal disorder caused by mutations in the gene cystic fibrosis transmembrane conductance regulator (CFTR), several peptides have been proposed as therapy, such as the S18 peptide as an ENaC antagonist [3], the alpha-1 proteinase inhibitor (PI) to reduce neutrophil elastase burden [4] or antimicrobial peptides (e.g., Esculentin [5], M33 [6] and Colistin [7]).

We and others have been working for many years on the development of therapeutic peptides acting as CFTR “stabilizers” [8,9,10,11,12]. Indeed, CFTR functional interactions at the apical membrane are regulated by the CAL (CFTR-associated ligand) protein, which limits cell surface levels of either wild-type CFTR or of the most common disease-associated mutant p.Phe508del-CFTR. CAL (also known as PIST, GOPC, and FIG) is a protein containing two coiled-coil domains and one PDZ domain interacting with the C-terminus of CFTR [13]. As shown by Guggino and co-workers, this CFTR–CAL interaction reduced the amount of surface CFTR through endocytic recycling [14], resulting in lysosomal degradation via the proteasome [15]. Based on this observation, selective inhibitors against the CFTR–CAL interaction should provide a novel class of CFTR “stabilizers”.

In line with this, we were the first to validate this approach by developing a peptide inhibitor of the CAL protein, iCAL36, which enhanced the CFTR membrane half-life in cultured airway epithelial cells and increased CFTR activity (+11%) [8,9]. Since then, other peptide-based inhibitors have been optimized in terms of their affinity [16], their proteolytic stability [10], or their cell-permeability [11].

Indeed, within the design of therapeutic peptides, the development of their optimal vectorization inside cells is a major point. Different vectors are used for the transfection of therapeutic peptides such as viral-based vectors, liposomes, nanoparticles, or cell-penetrating peptides (for reviews, see [17,18]). Cell-penetrating peptides (CPPs) were discovered thirty years ago, with the pioneering peptides Tat [19] and Penetratin [20]. To date, more than 100 different CPPs have been designed and constitute some of the most promising non-viral strategies for the delivery of peptides, proteins, and different nucleic acids [21]. They have gained increasing popularity because of their short length and efficient cell translocation ability, together with their cargoes.

Most CPPs are small, cationic peptides with typical lengths ranging from 5–30 residues, a median charge of +5, and an isoelectric point above 10 [22]. Although several different criteria have been proposed for the classification of CPPs based on their origin, sequence, function, or mechanism of uptake, no unified taxonomy of these peptides presently exists. Therefore, they can be simply categorized into three main classes according to their physico-chemical properties: cationic, amphipathic, and hydrophobic peptides [23]. Depending on the used CPP and even on the used cargo or cellular context, CPPs can pass through cell membranes via energy-dependent mechanisms or energy-independent ones without the implication of specific receptors [24].

In this study, we aimed to find out the most suitable CPP for the iCAL36 delivery in human epithelial cells. For this purpose, we selected several different CPPs, which were conjugated to the N-terminus of iCAL36. The cellular internalization (amount and localization) of these CPP-iCAL36 peptides was analyzed in Caco-2 and Calu-3 cells. Internalization mechanisms were elucidated using leakage assays, endocytosis inhibitors, endocytosis or vesicle markers, as well as via their internalization behavior in the presence of serum (model for mucus presence). Finally, we selected TatRI-iCAL36 as a more suitable CFTR stabilizer, showing for the first time an increase in p.Phe508del-CFTR after TatRI-iCAL36 treatment in epithelial 16HBE cells.

## 2. Materials and Methods

### 2.1. Peptide Synthesis

Peptide synthesis was performed using a LibertyBlue™ Microwave Peptide Synthesizer (CEM Corporation, Matthews, NC, USA) with an additional module of Discover™ (CEM Corporation, Matthews, NC, USA) combining microwave energy at 2450 MHz to the fluorenylmethoxycarbonyl (Fmoc)/tert-butyl (tBu) strategy. All chemicals or solvents were purchased from Sigma-Aldrich (Saint-Quentin-Fallavier, France) or Carlo Erba (Val-de-Reuil, France). Syntheses were conducted on a Fmoc-Ile-Wang resin (Bachem, Bubendorf, Switzerland, 0.25 mmol scale) using Fmoc-L- or Fmoc-D-amino acids (Bachem, Bubendorf, Switzerland). Tamra labeling of the peptide was performed by overnight coupling with 1.5 eq. 5(6)-carboxytetramethylrhodamine (Merck, Fontenay Sous Bois, France), 1.6 eq. HATU and 2 eq. DIEA (Merck, Fontenay Sous Bois, France) in DMF, then washing with DMF and CH_2_Cl_2_.

After purification by column chromatography, the peptide identity and purity were checked by LC-MS (Waters, Guyancourt, France). Unlabeled and Tamra-labeled peptides (Table 1) were used with a purity higher than 95%.

### 2.2. Cell Culture Conditions

Human epithelial colorectal adenocarcinoma (Caco-2, ATCC, HTB-37) and human epithelial pulmonary adenocarcinoma (Calu-3, ATCC, HTB55) cells were maintained in DMEM 4.5 g/L glucose supplemented with UltraGlutamine (Lonza, Levallois-Perret, France), 20% *v*/*v* fetal bovine serum (FBS from Thermo Fisher Scientific Inc., Rockford, IL, USA), 1% MEM non-essential amino acids, 1% penicillin/streptomycin, 1% sodium pyruvate and 1% sodium bicarbonate (all *v*/*v*, 100×, Thermo Fisher Scientific Inc., Rockford, IL, USA). Cells were passaged once a week using trypsin (0.05%, Life Technologies) and grown in a humidified incubator with 5% CO_2_ at 37 °C. All cells in experiments were used between passages 9 and 25 and were regularly tested for mycoplasma contamination. Cells were seeded at different densities (see below) for the corresponding experiments. In all cases, the medium was exchanged after 24 h, and cells were used 48 h after seeding.

Human bronchial epithelial cells 16HBEge-p.Phe508del (16HBE gene-edited CFTR p.Phe508del cells, generated by Dieter Grünert and distributed by the Cystic Fibrosis Foundation) were maintained in MEM medium supplemented with UltraGlutamine (Lonza, Levallois-Perret, France), 10% *v*/*v* fetal bovine serum (FBS from Thermo Fisher Scientific Inc., Rockford, IL, USA) and 1% *v*/*v* penicillin/streptomycin (100×, Life Technologies). Cells were passaged twice a week using trypsin (0.05%, Life Technologies) and grown in a humidified incubator with 5% CO_2_ at 37 °C. The 16HBE cells in experiments were used between passages 5 and 20 and were regularly tested for mycoplasma contamination. Cells were seeded at different densities (see below) for the corresponding experiments.

### 2.3. Cell Cytotoxicity Measurements

An evaluation of the cytotoxicity induced by CPP-iCAL36 conjugates was performed using a Cytotoxicity Detection KitPlus (LDH, (Merck, Fontenay Sous Bois, France) following the manufacturer’s instructions. In brief, 1.6 × 10^4^ cells (160 μL) were seeded in 96-well Nunc culture plates (Thermo Fisher Scientific Inc., Rockford, IL, USA). Then, 48 h post-seeding, cells were rinsed twice with D-PBS and then incubated with 160 μL of a 1 μM, 10 μM, or 100 μM CPP-iCAL36 solution (in triplicates) in OptiMEM (Life Technologies) for 24 h at 37 °C with 5% CO_2_. At least 3 wells were kept for the LDH negative control (0% viability) and for non-treated cells as a positive control (100% viability).

After a 24 h incubation period, negative controls were performed by adding Triton X-100 (Sigma-Aldrich, Saint-Quentin-Fallavier, France) (~10 min incubation at 37 °C). Afterward, 50 μL of supernatant was collected, mixed with 50 µL of the “dye solution/catalyst” mixture, and incubated in the darkness for 30 min at room temperature. The reaction was stopped by adding 25 µL of HCl (1 N) to each well before measuring the absorption at 490 nm. Relative toxicity (%) = ((exp. value − value non-treated cells)/(value triton − value non-treated cells)) × 100.

### 2.4. Cell Transfection Conditions for Flow Cytometry Acquisition

Here, 2 × 10^5^ cells (1 mL) were seeded in 24-well Nunc culture plates (Thermo Fisher Scientific Inc., Rockford, IL, USA). Next, 48 h post-seeding, cells were rinsed twice with D-PBS and then incubated with 500 μL of Tamra-CPP-iCAL36 solution at the indicated conditions.

***Standard transfection and kinetic measurements***. Cells were incubated for 1.5 h, 3 h, or 4.5 h with solutions of Tamra-CPP-iCAL36 (1 µM or 5 μM) in OptiMEM (500 µL) at 37 °C with 5% CO_2_.

***Transfection in the presence of serum***. Cells were incubated for 3 h with solutions of Tamra-CPP-iCAL36 at 1 μM or 5 μM in OptiMEM (500 µL) supplemented with 0%, 10% or 20% (*v*/*v*) FBS at 37 °C with 5% CO_2_.

***Transfection at 4 °C***. Cells were pre-incubated for 30 min at 4 °C (on ice) with OptiMEM, followed by incubation for 1.5 h at 4 °C with Tamra-CPP-iCAL36 (1 µM or 5 μM) in OptiMEM (500 µL).

***Transfection in the presence of endocytosis inhibitors or chloroquine***. Cells were pretreated for 30 min in OptiMEM (500 µL) using one of the following conditions: chlorpromazine (CPZ, 20 μM), nystatin (NYS, 50 μM), methyl-β-cyclodextrin (MBCD, 5 mM), sodium azide/2-deoxyglucose (NaN_3_/DG, 100 μM/60 μM) or chloroquine (CQ, 100 μM) at 37 °C with 5% CO_2_. Afterward, Tamra-CPP-iCAL36 (1 µM or 5 μM) in OptiMEM were added, and cells were incubated for a further 1.5 h at 37 °C with 5% CO_2_.

At the end of each incubation, cells were rinsed twice with D-PBS. To remove peptides sticking to the extracellular membrane, cells were treated for 10 min with 100 μL of trypsin (0.05%) at 37 °C with 5% CO_2_. After the addition of 400 μL of D-PBS supplemented with 5% (*v*/*v*) FBS, cells were transferred to 1.5 mL tubes and centrifuged (10 min, 4 °C, 1500 rpm). Cell pellets were resumed in 500 μL of D-PBS supplemented with 0.5% (*v*/*v*) FBS and 0.1% (*v*/*v*) 4′,6-diamidino-2-phenylindole (DAPI solution, Sigma-Aldrich, (Saint-Quentin-Fallavier, France), transferred to round-bottom polystyrene Falcon tubes (Thermo Fisher Scientific Inc., Rockford, IL, USA) and analyzed with a Fortessa LSR cytometer (Becton Dickinson, Pont-de-Claix, France; DAPI: λex = 405 nm/λem = 425–475 nm; TAMRA: λex = 561 nm/λem = 605–625 nm). Damaged cells were excluded with the DAPI labeling. For each condition, 10,000 events were recorded.

### 2.5. Cell Transfection Conditions for Confocal Microscopy Imaging

Here, 3 × 10^5^ cells (1 mL) were seeded in 35 mm FluoroDish culture dishes (WPI). Next, 48 h post-seeding, cells were rinsed twice with D-PBS and then incubated with 1 mL of Tamra-CPP-iCAL36 solution at the indicated concentration in OptiMEM at 37 °C with 5% CO_2_.

***Screening of Tamra-CPP-iCAL36 localization***. Cells were incubated for 3 h with solutions of Tamra-CPP-iCAL36 at 1 μM (P36, M36) or 5 μM (T36, dT36, TRI36) in OptiMEM at 37 °C with 5% CO_2_. Next, 10 min before the end of incubation, 10 μL of WGA-488 (final concentration: 1 μg/mL, Thermo Fisher Scientific Inc., Rockford, IL, USA) was added to label membranes.

***Transfection in the presence of endocytosis or lysosome markers***. Cells were incubated for 1.5 h with solutions of Tamra-TatRI-iCAL36 (TRI36, 5 μM) or Tamra-Pen-iCAL36 (P36, 1 μM), together with one of the following markers: Transferrin-Alexa488 (Transferrin-488, 50 μg/mL), cholera toxin subunit B-Alexa488 (CtB-488, 4 μg/mL), Dextran-pHrodo Green (Dextran-Green, 25 μg/mL) or LysoTracker™ Green DND-26 (1 μM) (all provided by Thermo Fisher Scientific Inc., Rockford, IL, USA) at 37 °C with 5% CO_2_.

***Transfection in the presence of endosome markers***. Cells (150,000 c/mL) were pretreated for 16 h before peptide incubation in a complete medium with one of the following markers: CellLight™ Early Endosomes-GFP, BacMam 2.0 (EE-GFP, 50 μg/mL) or CellLight™ Late Endosomes-GFP, BacMam 2.0 (LE-GFP, 50 μg/mL) (all supplied by Thermo Fisher Scientific Inc., Rockford, IL, USA). Afterward, cells were rinsed twice with D-PBS and then incubated with Tamra-CPP-iCAL36 solutions (TRI36 at 5 μM or P36 at 1 μM) in OptiMEM for 1.5 h at 37 °C with 5% CO_2_.

Next, 10 min before the end of each incubation, 10 μL of Hoechst 33342 (1 μg/mL, Sigma-Aldrich) to label the cell nuclei and 10 µL of WGA-Alexa488 (Alexa Fluor^TM^ 488 wheat germ agglutinin conjugate, 1 mg/mL, Thermo Fisher Scientific Inc., Rockford, IL, USA) to label cell membranes were added to the cell medium. Then, cells were rinsed twice with D-PBS and then covered with 1.5 mL of FluoroBrite (Life Technologies) for imaging using a Leica SP5-SMD confocal microscope (objective Leica HCS PL Apo CS 63x/1.4 NA oil objective; Hoechst 33342: λex = 405 nm/λem = 415–485 nm; TAMRA: λex = 561 nm/λem = 571–611 nm; Alexa488: λex = 488 nm/λem = 498–548 nm). After the acquisition, the images were processed under the same conditions and parameters using ImageJ software.

### 2.6. Cell Transfection Conditions for CFTR Quantification

***Transfection without VX pre-incubation***: Next, 3 × 10^5^ cells/well were seeded in collagen type I-coated 6-well Corning culture plates (Thermo Fisher Scientific Inc., Rockford, IL, USA). Then, 24 h post-seeding, cells were incubated with TatRI-iCAL36 diluted in serum-free medium at the indicated concentrations (1000 μL final volume). After 3.5 h of incubation, the medium was replaced with a fresh serum-supplemented medium containing VX-445 (3 µM final concentration, CliniSciences, Nanterre, France) and VX-809 (3 µM final concentration, CliniSciences, Nanterre, France), and cells were incubated for a further 24 h at 37 °C, 5% CO_2_.

***Transfection with VX pre-incubation***: Here, 2 × 10^5^ cells/well were seeded in collagen type I-coated 6-well Corning culture plates (Thermo Fisher Scientific Inc., Rockford, IL, USA). Then, 24 h post-seeding, cells were pre-incubated with a serum-supplemented medium containing VX-445/VX-809 (3 µM/3 µM final concentrations, CliniSciences, Nanterre, France). Again, 24 h later, the medium was replaced by TatRI-iCAL36 diluted in serum-free medium at the indicated concentrations (1000 μL final volume). After 3.5 h of incubation, the medium was again replaced with fresh serum-supplemented medium containing VX-445/VX-809 (3 µM/3 µM final concentrations), and cells were incubated for a further 24 h at 37 °C with 5% CO_2_.

***Western blot***: Cells transfected with TatRI-iCAL36 (with or without VX pre-incubation) were washed twice with PBS and lysed in 100 µL RIPA buffer (50 mM Tris pH 8.0, 150 mM sodium chloride, 1% Triton X-100, 0.1% SDS (sodium dodecyl sulfate, Sigma-Aldrich), including protease inhibitors (SigmaFAST, Sigma-Aldrich)). Protein concentrations were determined using the Pierce BCA Protein Assay (ThermoFisher). Cell extracts were separated by 7.5% Mini-PROTEAN^®^ TGX™ Precast Gel (Bio-Rad, Marnes-la-Coquette, France). After electrophoresis, samples were transferred onto Trans-Blot^®^ Turbo™ Mini PVDF Transfer membranes (Bio-Rad, Marnes-la-Coquette, France). As antibodies, we used anti-CFTR-450 or anti-CFTR-432 (recognizing both CFTR R domain, Cystic Fibrosis Foundation, Chapel Hill, NC, USA, 1:1000), anti-Lamin A/C (Sigma-Aldrich, 1:2000), and anti-mouse IgG HRP (Cell Signaling, Ozyme, Saint-Cyr-L’École, France, 1:2000). Blots were revealed with the Pierce ECL plus Western blotting substrate (Thermo Fisher Scientific Inc., Rockford, IL, USA) on an Amersham 600 imager (GE Healthcare Life Science, Vélizy-Villacoublay, France). The signal intensities of the blots were quantified using ImageJ software.

### 2.7. Statistical Analysis

One-way or two-way analysis of variance (ANOVA) with the Bonferroni test was used to compare data from control and multiple experimental groups. A confidence interval of 95% (*p* < 0.05) was considered statistically significant. Data analysis was performed with GraphPad Prism v.6.01 (GraphPad Software Inc., San Diego, CA, USA).

## 3. Results and Discussion

### 3.1. CPP Selection for iCAL36 Coupling and Evaluation of Their Effects on Cell Viability

In one of our past studies, we designed the iCAL36 peptide as a CFTR stabilizer [8]. This peptide, which was internalized via a lipid-based transfection reagent (BioPORTER^®^ QuikEase™ Protein Delivery Kit), induced an increase (+11%) in CFTR activity at the apical membrane of bronchial epithelia [9]. However, these results were obtained using a high iCAL36 concentration (500 µM), a value that is not compatible with a clinical application via pulmonary nebulization (equal to an extrapolated quantity of 6 g iCAL36 per day).

For that reason, we decided to couple the iCAL36 sequence with cell-penetrating peptides (CPPs) commonly used in the literature, namely Tat [19], MPG, and Penetratin (Pen), showing suitable internalization in different cell lines [26], as well as C6M1 and WRAP5 CPPs due to their secondary amphipathicity and their internalization via a direct cell membrane translocation [27,28] (Table 1).

Unfortunately, the first evaluations of C6M1-iCAL36 and WRAP5-iCAL36 revealed that both peptides were mainly stuck at or within the cell membrane, resulting in low cytoplasmatic release compared to Tat-iCAL36 and Pen-iCAL36 [25]. Therefore, both conjugates were excluded from this study.

To evaluate whether more protease-stable CPPs would increase cellular internalization, we included Tat analogs such as dTat (sequence with amino acids in isoform-D) [29] and TatRI (*retro-inverso* sequence in isoform-D) [30].

First of all, we evaluated the potential cytotoxic effects of iCAL36 (10 residues) and the five CPP-iCAL conjugates (23 to 37 residues) in human colorectal and human pulmonary epithelial cells (Caco-2 and Calu-3, respectively) using an LDH cytotoxicity kit. A screening of the literature revealed CPP concentrations varying from 1 μM to 50 μM, a range over which CPPs are normally not toxic [26,31,32]. Therefore, we evaluated the potent cytotoxic effects of the peptides by applying two commonly used concentrations (1 µM and 10 µM), as well as a more extreme concentration (100 µM) (Appendix A). The lower concentrations between 1 and 10 µM showed no significant effects on cell viability in either cell line compared to the non-treated cells (=100% cell viability). Only at the high CPP-iCAL36 concentration of 100 µM was reduced cell viability (~70–80%) observed for MPG-iCAL36, Pen-iCAL36, and dTat-iCAL36. These cytotoxic effects were not considered as a restriction for the screening of our peptides because (1) the goal of the CPP conjugation was the reduction in the applied iCAL36 concentration and (2) only peptide concentrations below 50 µM were to be used in the following experiments.

### 3.2. Comparison of CPP-iCAL36 Internalization and Cellular Localization

Tamra-iCAL36 and Tamra-CPP-iCAL36 peptides listed in Table 1 were tested in Caco-2 and Calu-3 cells, and cellular uptake was quantified by fluorescence measurements using flow cytometry (Figure 1A,B). In parallel, cell mortality was assessed using DAPI labeling. Compared to the cell viability assay (LDH, Appendix A), we observed that the procedure of cell preparation (trypsinization, washing, and transfer to the flow cytometer) reduced cell viability to ~80%, as shown for the non-treated cells. Therefore, we defined viable cells as cells having cell viability over 70% to evaluate the effect of the peptide incubation. As shown in Figure 1A,B, for all incubation conditions (1 µM or 5 µM) in both cell lines we did not observe cell viability below 70%.

Although all 5 CPPs were supposed from the literature to have cell-penetrating properties, they did not show the same degree of cellular internalization when conjugated to the iCAL36 peptide. However, compared to the naked iCAL36 peptide, all CPP-iCAL36 conjugates increased their cellular internalization. In both cell lines, MPG-iCAL36 and Pen-iCAL36 seemed to be superior to Tat-iCAL36 and its analogs.

To visualize the CPP-iCAL36 internalization and cellular localization, we performed confocal laser scanning microscopy (CLSM) measurements on living cells. Caco-2 or Calu-3 cells were incubated with the corresponding Tamra-CPP-iCAL36 peptides for 3 h. The incubation time was set from 1.5 h to 3 h to increase the fluorescence without changing the peptide concentrations as reported previously [28].

First, we performed control experiments without peptide (NT) or with Tamra-iCAL36 to show that no auto-fluorescence nor iCAL36 (without CPP) internalization was detected, respectively (Figure 1C,D). For the five Tamra-CPP-iCAL36 peptides, a similar cellular pattern was visible—they were all located as a punctuated pattern in the cytoplasms of both cell lines, with more signals in the Caco-2 cells compared to the Calu-3 cells, confirming the flow cytometry results (Figure 1C,D).

Due to the absence of net improvements compared to Tat-iCAL36, dTat-iCAL36 was excluded and we finally evaluated the time-dependent internalization of the four remaining CPP-iCAL36 conjugates in both cell lines. We determined that 30 min incubation was sufficient to measure significant cellular uptake. Longer incubation times (4.5 h) only increased the signal intensities by a factor of ~1.3 in Caco-2 cells. Only TatRI-iCAL36 revealed ~3-fold higher internalization (Appendix A). Calu-3 cells seemed to be more sensitive to the incubation duration with an increase in signal intensities by a factor of ~1.7–2.4 (Appendix A). Curiously, while it was not seen in Caco-2 cells, Tat-iCAL36 revealed a ~5.7-fold higher internalization compared to a factor of ~2.0–2.4 for the other CPP-iCAL36 peptides.

### 3.3. Internalization of the CPP-iCAL36 Peptides in the Presence of Serum

CF is characterized by a dysfunctional chloride CFTR channel, which leads to the production of a thick and viscous mucus layer perturbating the lung function of CF patients. The challenges in the development of CFTR stabilizers are related to the peptide retention within the mucus.

To evaluate this purpose in a simple cell model, CPP-iCAL36 transfections were performed in a serum-free medium (0%) or in a medium supplemented with 10% or 20% serum (all *v*/*v*) to mimic an environment rich in proteases and other serum proteins (Figure 2). As expected, neither the presence of the Tamra-CPP-iCAL36 peptides nor the increased serum concentrations in the transfection medium affected the cell viability of either tested cell line.

In contrast, the transfection efficiency of Tamra-CPP-iCAL36 peptides decreased in a dose-dependent manner with increasing serum concentrations. In Caco-2 cells, the transfection efficiency decreased by ~30% with a 10% serum incubation and by ~50% with a 20% serum condition for Tat-iCAL36, MPG-iCAL36, and Pen-iCAL36. As expected, due to the protease stability of TatRI, the peptide TatRI-iCAL36 revealed the lowest uptake changes (no significant difference with 10% serum and 18% decrease with 20% serum) (Figure 2A).

In Calu-3 cells, the observation was similar: we observed ~40% and ~50% decreases with 10% and 20% serum, respectively, for Pen-iCAL36; and ~20% and ~30% decreases with 10% and 20% serum, respectively, for Tat-iCAL36 and MPG-iCAL36 (Figure 2B). Here again, the TatRI-iCAL36 peptide showed nearly the same internalization behavior independent of the serum concentration (6% with 10% serum and 12% with 20% serum).

In conclusion, TatRI-iCAL36 was the conjugate most able to internalize in the presence of high serum content as compared to the others. Based on these results, we selected the two peptides TatRI-iCAL36 and Pen-iCAL36 to further evaluate their cellular internalization mechanism.

### 3.4. Evaluation of TatRI-iCAL36 and Pen-iCAL36 Membrane Interaction

CPPs are considered membrane-active peptides [33]. Some of these membrane-active peptides have a structural polymorphism (helicoidal structure formation in a specific environment) that drives their cellular uptake. For example, few residues were described to drive helical or beta secondary structures essential for membrane penetration in the presence of high detergent concentrations [34], at high peptide-to-lipid ratios, or in the presence of acidic phospholipids [35]. On the other hand, penetratin has been shown to be oriented parallel to the membrane surface [36].

To evaluate the structural features of the TatRI-iCAL36 and Pen-iCAL36 peptides, circular dichroism spectra (CD) were recorded for both peptides alone and compared to those measured in the presence of large unilamellar vesicles (LUVs) reflecting the lipid composition of cell membranes (for details see Appendix A). As expected, the signal compensation due to L- and D-isoform amino acid compositions of TatRI-iCAL provided a weak CD profile that could not be associated with any specific structure. In addition, no significant change was detected in the presence of increasing LUV amounts or Trifluoroethanol (TFE), a solvent favoring helix formation (Appendix A). For the Pen-iCAL36 peptide, we observed a random-coiled spectrum characterized by a minimum at 198 nm (Appendix A). No obvious changes were recorded after the addition of LUVs, even if an α-helical conformation could be forced in the presence of TFE. However, the latter has no pharmacological consequences for the Pen-iCAL36 peptide, as TFE is only used as a conformational positive control.

To further evaluate the potential membrane-active properties of both peptides, we performed leakage experiments on LUVs encapsulating a quenched fluorescent dye incubated with increasing peptide concentrations (for protocol details see Appendix A). In a previous study, we used this model to demonstrate the membrane activity of WRAP5-iCAL36, showing leakage of 85% [37]. However, for both peptides analyzed here, no significant LUV leakage was detected, even at high peptide concentrations (Data not shown).

Altogether, these results suggested that TatRI-iCAL36 and Pen-iCAL36 were not able to adopt a particular secondary structure (CD), resulting in the absence of membrane destabilization properties (no leakage). Both CPP-iCAL36 peptides are probably internalized via endocytosis-dependent pathways.

### 3.5. Dissecting the Internalization Mechanism of TatRI-iCAL36 and Pen-iCAL36

Since the discovery of CPPs, hundreds of studies have been performed to identify the mechanism(s) of cellular entry [38,39], which occurs mainly via endocytosis-dependent pathways (macropinocytosis, clathrin- and caveolin-dependent endocytosis) [40,41,42], even if in some cases direct cell membrane translocation has been reported at high CPP concentrations [40,43].

If an endocytosis-dependent pathway was activated for the peptide internalization, the fluorescent-labeled peptides would be entrapped in endosomes, where the fluorescent dye would be mainly quenched [25]. To confirm this hypothesis, we incubated the Caco-2 cells with an endosomolytic agent Chloroquine (CQ) to determine if we would obtain an increased fluorescent signal (Figure 3A). First of all, we confirmed that CQ treatment did not induce significant cytotoxicity compared to untreated Caco-2 cells (80 ± 10%). Then, we showed that the addition of CQ doubled the intracellular fluorescence signal of Tamra-TatRI-iCAL36 and Tamra-Pen-iCAL36, demonstrating the release of peptides from endosomes. These results confirmed the involvement of an endocytosis mechanism and indicated that non-negligible fractions of the peptides were trapped in endosomes.

Because endocytosis-dependent internalization requires energy, we downregulated the energy-dependent pathways by incubating cells either at 4 °C or under ATP depletion (NaN_3_/_2_-Deoxy-Glucose at 37 °C) to compare the Tamra-TatRI-iCAL36 and Tamra-Pen-iCAL36 internalization efficiency to the standard 37 °C incubation condition. The internalization of both peptides was reduced under ATP depletion (55% of the signal at 37 °C) and was almost lost at 4 °C (10–15% of the signal at 37 °C), confirming an endocytosis-dependent internalization (Figure 3B).

To further depict the exact internalization mechanism, we evaluated the cellular uptake of both Tamra-TatRI-iCAL36 and Tamra-Pen-iCAL36 peptides in the presence of specific chemical inhibitors of different endocytosis pathways such as chlorpromazine (CPZ, clathrin-dependent endocytosis inhibitor), nystatin (NYS, disrupting caveolar structure and function) and methyl-β-cyclodextrin (MBCD, lipid raft inhibitor) (for a review see [43]). Even if in the CPP field all of these chemical inhibitors were commonly used in several investigations to hamper endocytic processes, the maximal concentration with low cytotoxicity for each inhibitor was first determined to ensure maximal inhibition (data not shown). As expected, treatment of Caco-2 cells with CPZ, NYS or MBCD induced slight cytotoxicity (5–20% loss of viability), as observed for ATP depletion compared to non-treated cells (80 ± 10%) (Figure 3B). Curiously, endocytosis inhibitors did not lead to a strong change in the intracellular fluorescence of Tamra-TatRI-iCAL36–only slight non-significant decreases (21%) were observed in the presence of NYS and MBCD. Nearly the same results were observed for the intracellular fluorescence of Tamra-Pen-iCAL36 but with a significant reduction (28%, **p*) in internalization in the presence of NYS, revealing the implication of caveolae-dependent endocytosis. These results suggested an internalization mechanism depending on multiple endocytosis-dependent pathways.

Because a number of these chemical endocytosis inhibitors lack specificity, it was possible that they could give misleading results [44,45]. Therefore, we further evaluated the internalization of both peptide conjugates by confocal microscopy using three specific endocytosis markers, namely transferrin, cholera toxin subunit B (CtB), and dextran, to visualize the clathrin-dependent, caveolin-mediated and macropinocytotic pathways, respectively [46]. As observed in Figure 3C,D, colocalization of Tamra-TatRI-iCAL36 and Tamra-Pen-iCAL36 peptides was relatively strong with all endocytosis markers (see arrows). These results were surprising because previous analyses with endocytosis inhibitors showed nearly no significant effect on internalization. On the other hand, the results were coherent with a mechanism involving multiple replaceable endocytosis-dependent pathways.

### 3.6. Intracellular Fate of TatRI-iCAL36 and Pen-iCAL36

To complete the study of the internalization mechanism of both Tamra-TatRI-iCAL36 and Tamra-Pen-iCAL36 peptides, we investigated their fate once inside the cell. A compound taken up by endocytosis will be entrapped in endosomes (called early endosomes). After a pH drop, early endosomes will be changed into late endosomes and then merge into lysosomes, where the peptides will be recycled or degraded.

To follow vesicle trafficking, we performed colocalization experiments between Tamra-TatRI-iCAL36 and Tamra-Pen-iCAL36 peptides and between specific markers of early and late endosomes, as well as lysosomes. As shown in Figure 4A,B, neither peptide fully colocalized with early endosome markers. Instead, they showed strong degrees of colocalization with late endosomes and with lysosomes (see arrows). These results suggested that after 1.5 h of incubation, the majority of the peptide internalized by endocytosis was found in late endosomes or lysosomes, confirming the internalization via endocytosis-dependent pathways.

In conclusion, we could deduce that both CPP-iCAL36 peptides were internalized via an energy-dependent endocytosis mechanism, which could include multiple endocytosis-dependent pathways.

### 3.7. Stabilizing Effect of TatRI-iCAL36 Increases CFTR Amount

Comparing the internalization properties of Tamra-TatRI-iCAL36 and Tamra-Pen-iCAL36, we could not determine an overall better candidate between both peptides. However, because TatRI-iCAL36 internalization was less reduced in the presence of serum (Figure 2A,B) and because its sequence is shorter than Pen-iCAL36 (Table 1), we selected it for activity measurement. Indeed, the measured inhibition constant (Ki) revealed a specific interaction of Tat-RI-iCAL36 with the CAL PDZ domain, with a value of 1.5 ± 0.4 µM. More importantly, interactions of the peptide with the PDZ domains of the NHERF2 proteins (N2P1 and N2P2) were not observed (Ki > 5000 µM) (for protocol details see the Appendix A), as the NHERF2-CFTR interaction was reported to stabilize the CFTR at the plasma membrane of epithelial cells [47].

Afterward, we first evaluated the internalization properties of Tamra-TatRI-iCAL36 in a human bronchial epithelial gene-edited CFTR p.Phe508del cell line (16HBEge-pPhe508del, noted 16HBEge) as a more appropriated model for the CF pathology. As shown in Figure 5A, we confirmed via confocal microscopy that the Tamra-labeled iCAL36 peptide was not internalized, and more importantly that the Tamra-TatRI-iCAL36 peptide revealed an impressive internalization profile in all cells after a 3.5 h transfection. Additionally, we confirmed that at the used concentrations (between 1 µM and 10 µM), no cytotoxic effects were induced by the peptide incubation (LDH and Cristal Violet assays, Appendix A).

The stabilizing property of TatRI-iCAL36 was evaluated indirectly. Because the peptide might inhibit the interaction of the CAL protein with the C-terminus of the mutated p.Phe508del-CFTR, CFTR degradation via the endosomal–lysosomal pathway should be reduced or abolished. As a consequence, an increase in the total CFTR amount should be detected in the 16HBEge cells. These experiments were performed in the presence of two CFTR correctors VX-445 (elexacaftor) and VX-809 (lumacaftor), which were found to be particularly effective when administrated together without significant cytotoxic effects [48]. Furthermore, a VX cocktail was always used during these experiments because p.Phe508del-CFTR can only be quantified if enough folded receptors reach the membrane.

First, we confirmed that the VX-445/VX-809 corrector cocktails increase the CFTR amount in the 16HBEge cells after 24 h incubation. Indeed, as shown in Figure 5B, we observed a 3.5-fold increase in p.Phe508del-CFTR. Curiously, using these VX-incubation conditions, we could not observe an additive effect of the TatRI-iCAL36 peptide.

We thought that the lack of peptide effect could be due to the VX incubation conditions, which were only performed after peptide incubation. The stabilizing properties of the TatRI-iCAL36 would only be observed if the correctors have induced enough p.Phe508del-CFTR folding for proper accumulation at the apical membrane of epithelial cells. Therefore, we decided to perform pre-peptide incubation with the VX-cocktail to ensure that p.Phe508del-CFTR avoided proteasomal degradation and reached the plasma membrane. Using this new incubation condition, we revealed a 3-fold increase in the CFTR amount in the 16HBEge cells after VX treatment alone. More interestingly, we observed for the first time 4-fold and 6-fold increases in the p.Phe508del-CFTR quantity after 5 µM and 10 µM peptide incubation, respectively (Figure 5C). This effect reached a plateau when we increased the TatRI-iCAL36 peptide concentration to 20 µM.

At this stage, these results are very promising in the development of TatRI-iCAL36 as a new CF treatment in combination with VX-445/VX-809 correctors.

## 4. Discussion

Therapeutic peptides, which address unmet medical needs in complement to low molecular weight drugs, have seen renewed interest in recent years. For the development of therapeutics against cystic fibrosis (CF), different modulators of the mutated chloride ion channel CFTR were developed and approved by the Food and Drug Agency (FDA), such as correctors (Lumicaftor, Elexacaftor) or potentiators (Ivacaftor). In parallel, CFTR--stabilizing peptides were also developed by our group [8,9,12,25] and by others [11,16,49] to further increase the CFTR amount in epithelial cells by decoupling the negatively regulating CFTR–CAL interaction.

In a previous study, we engineered iCAL36 as a CFTR stabilizer [8] and validated its CFTR-stabilizing effect using a lipid-based transfection [9]. To reduce the applied iCAL36 quantity (500 µM) for a more physiological application, we conjugated the iCAL36 stabilizer to five different cell-penetrating peptides (CPPs): Tat and its analogs dTat (D-isoform) and TatRI (*retro-inverso*), as well as MPG and Penetratin based on a previous CPP screen [25]. To determine the optimal CPP for iCAL36 transfection, we performed a precise screening of the internalization properties (depending on their concentrations, incubation time, cellular localization, and presence of serum) of these five CPP-iCAL36 peptides. All candidates were able to be internalized in the cytosol of epithelial cells compared to iCAL36 alone, even if the transfection ratios were different between the peptides. Based on their capacity to internalize in the presence of serum, TatRI-iCAL36 and Pen-iCAL36 were pre-selected to evaluate their cellular uptake mechanism (using endocytosis inhibitors, endocytosis, and vesicle markers). The internalization of both conjugates occurred mainly through energy-dependent endocytosis.

Based on the fact that TatRI-iCAL36 had a shorter sequence and was easier to synthesize, we selected this peptide as the more appropriate CPP for iCAL36 transfection. Indeed, TatRI-iCAL36 had a remarkable affinity to CAL (~1 µM) and an impressive selectivity (no interaction with the NHERF PDZ domains) compared to the iCAL36 peptide (22 µM) or other published iCAL36 analogs (~11 µM) [50]. More importantly, compared to the previously used iCAL36 peptide transfected by a lipid-based reagent, we could reduce the peptide concentration from 500 µM to 10 µM during the experiments performed to measure CFTR stabilization. Using this concentration, and in the presence of the correctors VX-445 and VX-809, we revealed a significantly increased amount of p.Phe508del-CFTR in bronchial epithelial cells. In turn, this would reduce the theoretical extrapolated dose for the patients from 6 g/day to approximately 0.3 g/day.

Taken together, the knowledge provided by this report will significantly contribute to pushing forward the development of novel CFTR stabilizers as co-therapies to the currently used Trikafta^®^ treatment (elexacaftor VX-445/tezacaftor VX-661/ivacaftor VX-770).

## Figures and Tables

**Figure 1 pharmaceutics-14-00808-f001:**
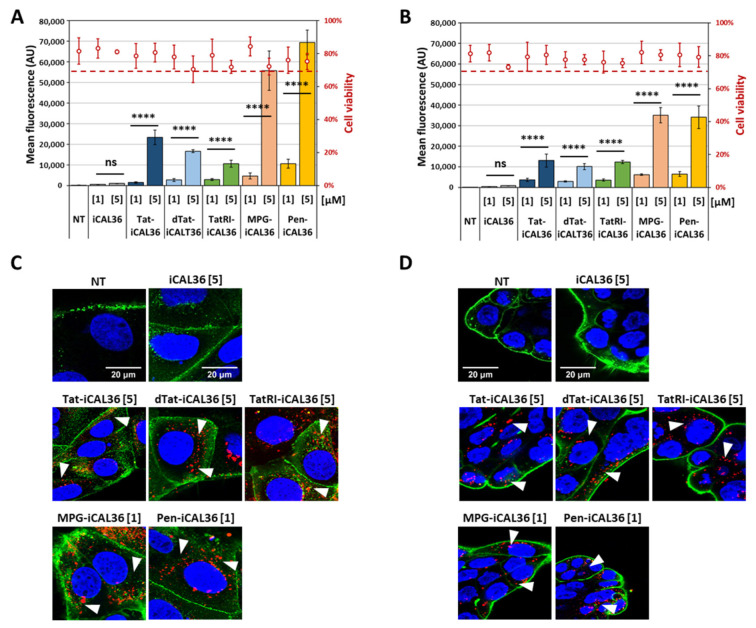
Comparison of CPP-iCAL36 internalization and cellular localization. (**A**) Caco-2 and (**B**) Calu-3 cells were incubated with Tamra-iCAL36 or Tamra-CPP-iCAL36 solutions (1 μM = [1] or 5 µM = [5]) in OptiMEM for 1.5 h and compared to non-treated cells (NT). After cell trypsinization (to remove membrane externally bound peptides), fluorescence and cell viability were assessed via flow cytometry. Graphical representation of data obtained from four independent measurements (mean ± SD, n = 4). A statistically significant difference was noticed between the two different used concentrations for all CPP-iCAL36 peptides (one-way ANOVA with Bonferroni test, statistical relevance given as **** *p* < 0.0001 and ns > 0.05). Living (**C**) Caco-2 and (**D**) Calu-3 cells were imaged by confocal laser scanning microscopy (CLSM) after 3 h incubation with the Tamra-iCAL36 and Tamra-CPP-iCAL36 peptides (red, see white arrows) at the indicated concentrations (1 µM = [1] or 5 µM = [5]) in OptiMEM. Next, 10 min before the end of the incubation, Hoechst dye (blue = nuclei) and WGA-Alexa488 (green = cell membrane) were added. Cells were washed and covered with DMEM FluoroBrite before imaging. White bars represent 20 μm. The results of the Tat-iCAL36 (1 µM = [1]) and MPG-iCAL36 (1 µM = [1]) peptides on Caco-2 cells were previously reported [25].

**Figure 2 pharmaceutics-14-00808-f002:**
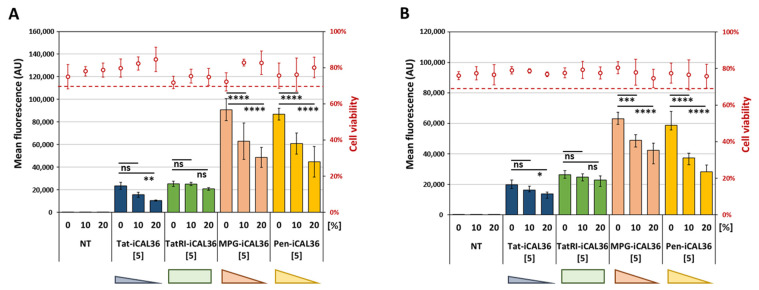
Internalization efficiency of CPP-iCAL36 peptides in the presence of serum. Caco-2 (**A**) and Calu-3 (**B**) cells were incubated with the Tamra-CPP-iCAL36 peptides (5 µM = [5]) for 3 h and compared to non-treated cells (NT) at the indicated serum concentration (0%, 10% or 20% FBS). After cell trypsinization (to remove membrane externally bound peptides), fluorescence and cell viability were assessed via flow cytometry. Graphical representation of data obtained from four independent measurements (mean ± SD, n = 4). A statistically significant difference was noticed between the three different used serum percentages for MPG-iCAL36 and Pen-iCAL36, whereas no statistical difference was revealed for TatRI-iCAL36 (2-way ANOVA with Bonferroni test, statistical relevance given as **** *p* < 0.0001, *** *p* < 0.001, ** *p* < 0.01, * *p* < 0.05, ns > 0.05).

**Figure 3 pharmaceutics-14-00808-f003:**
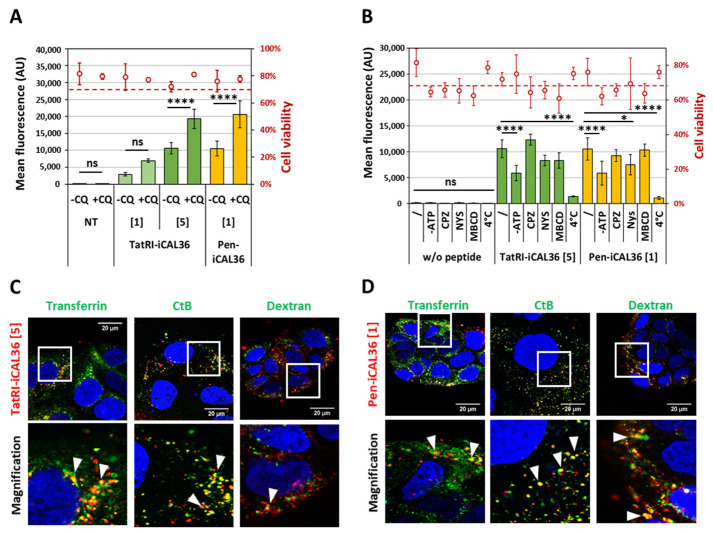
Dissecting the internalization mechanism of TatRI-iCAL36 and Pen-iCAL36. (**A**) Caco-2 cells were incubated with Tamra-TatRI-iCAL36 (1 µM = [1] or 5 µM) = [5] and Tamra-Pen-iCAL36 (1 µM = [1]) and compared to non-treated cells (NT) in the absence or presence of chloroquine (CQ, 100 µM). After cell trypsinization (to remove external peptides), the fluorescence and cell viability were assessed via flow cytometry. Graphical representation of data obtained from four independent measurements (mean ± SD, n = 4). A statistically significant difference was noticed between the incubation conditions without and with CQ for TatRI-iCAL36 and Pen-iCAL36 at 5 µM, whereas no statistical difference was revealed for TatRI-iCAL36 at 1 µM (one-way ANOVA with Bonferroni test, statistical relevance given as **** *p* < 0.0001 and ns > 0.05). (**B**) Caco-2 cells were incubated with Tamra-TatRI-iCAL36 (5 µM) and Tamra-Pen-iCAL36 (1 µM) at 4 °C or in the presence of either ATP-depleting reagents (NaN_3_/DG) or endocytosis inhibitors (CPZ, NYS or MBCD) at 37 °C. The same conditions were used on cells without peptides (w/o peptide) to determine the effect of the endocytosis inhibition. For all conditions, fluorescence and cell viability were acquired via flow cytometry after cell trypsinization (to remove membrane externally bound peptides). Graphical representation of data obtained from four independent measurements (mean ± SD, n = 4). A statistically significant difference compared to untreated condition was noticed for ATP depletion and 4 °C incubation for TatRI-iCAL36 and Pen-iCAL36 (2-way ANOVA with Bonferroni test, statistical relevance given as * *p* < 0.05, **** *p* < 0.0001 and ns > 0.05). (**C**,**D**) Caco-2 cells were incubated with Tamra-TatRI-iCAL36 (5 µM = [5]) and Tamra-Pen-iCAL36 (1 µM = [1]) in the presence of endocytosis markers such as transferrin, cholera toxin subunit B (CtB) or Dextran. Next, 10 min before the end of the incubation, Hoechst dye (blue = nuclei) and WGA-Alexa488 (green = cell membrane) were added. Cells were washed and covered with DMEM FluoroBrite before imaging. White bars represent 20 μm.

**Figure 4 pharmaceutics-14-00808-f004:**
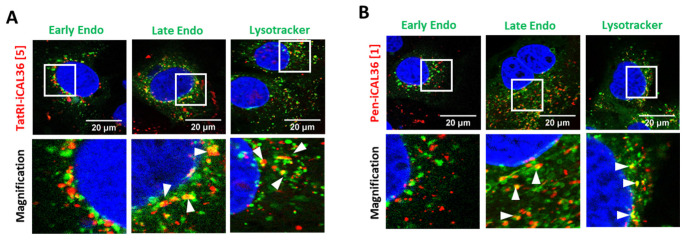
Dissecting vesicular entrapment of TatRI-iCAL36 and Pen-iCAL36. Caco-2 cells were incubated with Tamra-TatRI-iCAL36 (5 µM = [5] (**A**)) and Tamra-Pen-iCAL36 (1 µM = [1] (**B**)) in the presence of vesicle markers such as early and late endosomes, as well as lysosomes. Here, 10 min before the end of the incubation, Hoechst dye (blue = nuclei) and WGA-Alexa488 (green = cell membrane) were added. Cells were washed and covered with DMEM FluoroBrite before imaging. White bars represent 20 μm.

**Figure 5 pharmaceutics-14-00808-f005:**
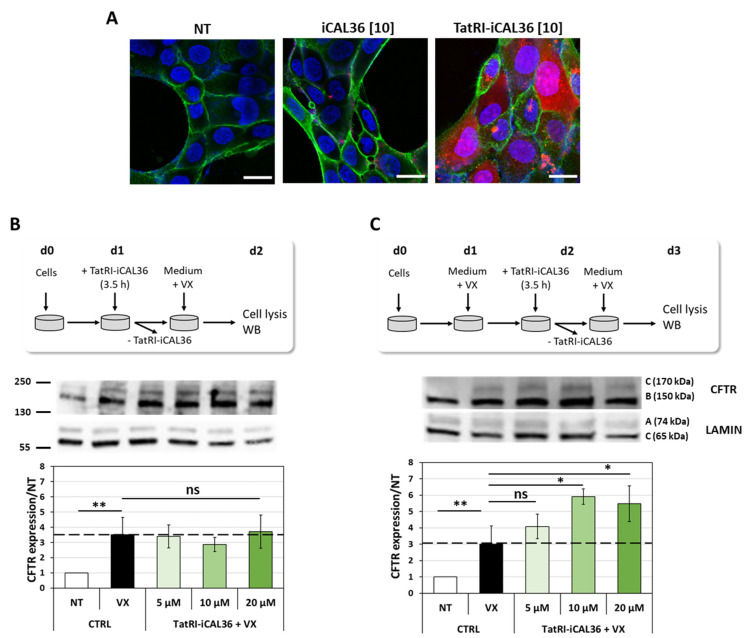
Dissecting the internalization and activity of TatRI-iCAL36 in 16HBEge cells. (**A**) Living 16HBEge cells were imaged by CLSM after 3 h incubation with Tamra-iCAL36 (10 µM = [1]) or Tamra-TatRI-iCAL36 (10 µM = [10])) in OptiMEM. Then, 10 min before the end of the incubation, Hoechst dye (blue) and WGA-Alexa488 (green) were added. Cells were washed and covered with DMEM FluoroBrite before imaging. White bars represent 10 μm. (**B**) The 16HBEge cells were incubated with TatRI-iCAL36 (5 µM, 10 µM, or 20 µM) in the presence of the corrector VX-445/VX-809 cocktail as indicated in the scheme. On day 2 (d2), the cells were lysed and p.Phe508del-CFTR was quantified by Western blot. Graphical representation of data obtained from three independent Western blot quantifications (mean ± SD, n = 3). A statistically significant difference was noted between non-treated (NT) and VX-445/VX-809-treated (VX) cells, whereas no statistical significance was revealed for the TatRI-iCAL incubation (one-way ANOVA with Bonferroni test, statistical relevance given as ** *p* < 0.01 and ns > 0.05). (**C**) The 16HBEge cells were pre-incubated with the corrected VX-445/VX-809 for 24 h. Afterward, the cells were incubated with TatRI-iCAL36 (5 µM, 10 µM, or 20 µM) in the presence of the VX-445/VX-809 corrector cocktail as indicated in the scheme. On day 3 (d3), cells were lysed and p.Phe508del-CFTR was quantified by Western blot. Graphical representation of data obtained from three independent Western blot quantifications (mean ± SD, n = 3). A statistically significant difference was noticed between non-treated (NT) and VX-445/VX-809-treated (VX) cells, as well as between the VX-treated and the TatRI-iCAL36-treated cells at 10 µM and 20 µM (one-way ANOVA with Bonferroni test, statistical relevance given as ** *p* < 0.01, * *p* < 0.05 and ns > 0.05).

**Table 1 pharmaceutics-14-00808-t001:** Peptide sequences used in this study.

Name	Code	Sequence	Residues	pI
iCAL36	36	ANSRWPTSII	10	9.79
Tat-iCAL36 ^1^	T36	GRKKRRQRRRPPQ-ANSRWPTSII	23	12.78
dTat-iCAL36	dT36	grkkrrqrrrppq-ANSRWPTSII	23	12.78
TatRI-iCAL36	TRI36	qpprrrqrrkkrg-ANSRWPTSII	23	12.78
Pen-iCAL36	P36	RQILIWFQNRRMKWKK-ANSRWPTSII	26	12.48
MPG-iCAL36 ^1^	M36	GALFLGWLGAAGSTMGAWSQPKKKRKV-ANSRWPTSII	37	12.03

^1^ Sequences presented in Seisel et al. 2019 [25]; pI was determined using the Expasy ProtParam tool.

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
