# Peer review of "Highway to Cell: Selection of the Best Cell-Penetrating Peptide to Internalize the CFTR-Stabilizing iCAL36 Peptide"

_pharmaceutics, 2022, doi:10.3390/pharmaceutics14040808_

Round 1

Reviewer 1 Report

The manuscript by Quentin et al optimized a CFTR-stabilizing iCAL36 peptide based on their previous studies. By coupling with different cell-penetrating peptides, they generated 6 different candidates. Together with cell cytotoxicity measurements, internalization assays and CFTR quantification, they identified TatRI-iCAL36 is the best candidate which can significantly increase CFTR amount in combination with VX cocktails. Overall, these studies have far-reaching significance for the development of a new CF treatment. I am in principle very supportive of this manuscript, which I believe can be improved following input from the authors as requested below:

  1. Fig 1C and 1D, are the images in the same magnification? If yes, why do their nuclear appear in different sizes (e.g. T36 in 1C is smaller than others and dT36 in 1D is larger than others)? If not, please label the scale bar in each image.
  2. There are no figure legends for all the supplementary figures.
  3. Please label A and B in the Figure S2.
  4. When the authors investigated the internalization mechanism of CCP-iCAL36, the positive controls are needed for the inhibitor treatments. The convinced results for different inhibitors treatments should be that the CCP-iCAL36 can enter the cells while the endocytosis markers (or other positive controls) stay in the cell surface.
  5. As the authors treated the 16HBEge cells with both CCP-iCAL36 and VX cocktails, how is the cytotoxic effect of both CCP-iCAL36 and VX cocktails treatments?

Reviewer 2 Report

The manuscript under review: “Highway to cell: Selection of the best cell-penetrating peptide to internalize the CFTR stabilizing iCAL36 peptide” by Seisel Quentin, Lakumpa Israpong, Josse Emilie, Vivès Eric, Varilh Jessica, Taulan-Cadars Magali and Boisguérin Prisca, is devoted to selection of the best cell-penetrating peptide in order to internalize the CFTR stabilizing iCAL36 peptide. The authors performed a lot of studies and obtained a promising candidate peptide which showed an additive increase of CFTR amount in the presence of VX-445/VX-809 compared to VX-445/VX-809 treatment alone on CFTR p.Phe508del cells. The study is well-designed and with no doubts has new interesting data. The study meets the scope of the Journal. Current manuscript under review can be published, however, several points should be addressed first.

  1. Line 72. Median isoelectric point (pI) of the peptides should also be indicated, as pI is more relevant for the physiological conditions than overall charge.
  2. Why the authors did not use polarized monolayers of Caco-2 and Calu-3 cells instead of single cells? It is known that the most advantageous property of that cell lines is the ability to spontaneously polarized after reaching the confluent monolayer. After polarization, the cells form tight junctions, express apical enzymes, and the monolayers form characteristic apical brush borders with microvilli. Unpolarized single cells may have different properties than polarized cells in monolayer.
  3. The authors gave the definitions of the codes used in the manuscript for the cell-penetrating peptides as follows:

iCAL36 for 36

Tat-iCAL36 for T36

dTat-iCAL36 for dT36

TatRI-iCAL36 for TRI36

Pen-iCAL36 for P36

MPG-iCAL36 for M36,

however, these short codes appear only in Figures, while the full abbreviations of the peptides, like Tat-iCAL36, are given in the text flow of the manuscript. It is confusing, because it is hard for the reader to memorize both writing form for each peptide, particularly when ”decoding” Table 1 is located only on page 6. The authors should replace short codes on figures by their full abbreviations.

  1. Line 176. There is no mention in whole manuscript about the target of WGA-488 dye.
  2. Lines 222-223. What is the difference between anti-CFTR-450 and anti-CFTR-432? There is no explanation.
  3. Line 261. Pen_iCAL36 should be Pen-iCAL36.
  4. Line 229,236,263. The term “vectorization” is quite not appropriate here, as the term “vector” is commonly applied to nanoparticles and liposomes, which act as vectors in the field of drug delivery:

Line 229. “CPP selection for iCAL36 vectorization…” should be        replaced by ”CPP selection for iCAL36 coupling…”

Line 236. “…we decided to vectorize the iCAL36 sequence              with…“ should be “…we decided to couple the iCAL36 sequence with…“, and so on.

  1. Line 233. Epitheliums should be epithelia (plural form).
  2. Line 300. NT – what it means? Please, add the definition in all figure legends.
  3. Line 380. “here, we were not able to detect…“ I suppose should be “here, a significant LUV leakage was not detected even at high peptide concentrations“?
  4. Line 453. The word evolution in the name of section 3.5 should be replaced by the word fate, for instance.
  5. Figure S2 in Supplementary file. Please, provide in the figure caption the method used. As I understand, the results were obtained with flow cytometry? Also [μM] should be added on the figure.
  6. Line 370. I would recommend to replace Pen-iCAL36 conjugate to Pen-iCAL36 hybrid peptide or just Pen-iCAL36 peptide, as term conjugate is commonly used for proteins or peptides conjugated with some marker molecules, like fluorochrome (f.e. Tamra), or functional enzymes. If Pen-iCAL36 conjugate means here unlabelled Pen-iCAL36 peptide. The same is applicable to line 242. Or is this term used for Tamra-conjugated hybrid peptides? The authors should clarify this within whole manuscript. Also in abstract (line 24).
  7. The authors observed changes in CD spectra of Pen-iCAL36 peptide in the presence of TFE. Does it have any application for pharmacology? Which consequences this feature of Pen-iCAL36 might have?
  8. Section 5. Conclusions should not have any references (lines 550, 554, 558, 571) and new data, which were not discussed in previous sections (lines 569-571), as Concluding paragraphs should be clear and sum up what have been presented in the research without sounding redundant.
  9. Line 233-234. It was stated that, according to the previous work, iCAL36 peptide alone is active in quite high concentration of 500 μM, which is not compatible with a clinical application (equal to 6 g iCAL36 per day). The authors should add into the discussion the same evaluation for clinical application of candidate TatRI-iCAL36 hybrid peptide.
  10. There is no statistics in current research. The authors should add it. Please, provide the level of statistical significance (p-value) for each figure with error bars (Fig 1A,B; Fig 2A,B; Fig 3A,B; Fig 5B,C; Fig S1; Fig S2). p-value should reflect significance of the differences between control (NT) and experimental sample groups.

Reviewer 3 Report

Dear Author,

The research article entitled “Highway to cell: Selection of the best cell-penetrating peptide to internalize the CFTR stabilizing iCAL36 peptide” has been intensively reviewed and evaluated. A literature review shows that this was an enrichment and improvement of previous studies. According to the data given in the text, it was found to be newsworthy and added valuable information to the literature. Consequently, the manuscript could be accepted after minor revisions.

Hereby, I would like to present my suggestions and revisions.

  1. Revision_1 (line_234): Could you please indicate the route of administration (such as; i.v. infusion, i.m. application, etc.) for the clinical subjects.
  2. Revision_2 (line_255): Reference 31 is not compatible with the specified range. The range or the literature data might be mixed. Please check and revise.

Sincerely

Reviewer 4 Report

The manuscript entitled "Highway to cell: Selection of the best cell-penetrating peptide to internalize the CFTR stabilizing iCAL36 peptide" describes the conjugation CFTR stabilizing iCAL36 with various cell-penetrating peptides (CPP), to identify suitable CPP-peptide conjugate which can facilitate delivery of active peptide in the cells. The strength of the paper is that the authors used various CPP-peptide conjugates and perform analyses on several aspects like, internalization of conjugates, cellular toxicity and activity of the peptides. The MS can be accepted for publication after taking care of minor concerns and questions described below:

1) Page 1, Line 14-17: Simplify and rephrase this statement

2) Page 1, Line 21: MPG "or" Penetration. I think the right word here will be "and" instead of "or"

3) Page 1, Line 34-36: Rephrase and Simplify this statement 

4) Page 2, Line 46: Insert comma after the peptide name

5) Page 9, Line 373: Spectra instead of spectrum and maxima instead of maximum

6)Page 9, Line 379-380: Is there any associated data (figures or tables) mentioned in the main text or supplementary files. if not than it should be mentioned.

7) Page 11, Line 448: Did the authors tried combination of two or three endocytosis inhibitors to pinpoint the exact membrane internalization pathway for the CPP-inhibitors conjugate tested? 

8) Page 12, Line 484-488: Is there any associated data (figures or tables) mentioned in the main text or supplementary files. if not than it should be mentioned.

9) Page 13, Line 497: spell check for cristal

10) Page 13, Line 504: Briefly describe the role of CFTR correctors here. It would be useful for the naive readers

11) Page 14, Line 525: mentioned 24 hours instead of just 24

Round 2

Reviewer 2 Report

In current version of the manuscript the shortened codes of the hybrid peptides (P36, M36, T36, dT36, TRI36) are presented only in two places throughout whole manuscript - in line 175 and Table 1, so, they can be substituted by full abbreviations in line 175 and be omitted in Table 1. I thank the authors for clearly and thoroughly explaining of my questions and remarks. The authors have improved the revised manuscript greatly on several critical points, which have enormously improved the readability of the manuscript and clarity of the results.